# The Community Structure of eDNA in the Los Angeles River Reveals an Altered Nitrogen Cycle at Impervious Sites

Savanah Senn [1,2,*], Sharmodeep Bhattacharyya [1,3], Gerald Presley [1,4], Anne E. Taylor [5], Rayne Stanis [2], Kelly Pangell [2], Daila Melendez [2,5] and Jillian Ford [2]

1    Environmental Sciences Graduate Program, Oregon State University, Corvallis, OR 97331, USA; sharmodeep.bhattacharyya@oregonstate.edu (S.B.); gerald.presley@oregonstate.edu (G.P.)
2    Department of Agriculture Sciences, Los Angeles Pierce College, 6201 Winnetka Avenue, PMB 553, Woodland Hills, CA 91304, USA; raynelace@gmail.com (R.S.); pangelkl4875@student.laccd.edu (K.P.); waggond1440@student.laccd.edu (D.M.); fordjm7164@student.laccd.edu (J.F.)
3    Department of Statistics, Oregon State University, Corvallis, OR 97331, USA
4    Department of Wood Science & Engineering, Oregon State University, Corvallis, OR 97331, USA
5    Department of Crop and Soil Sciences, Oregon State University, Corvallis, OR 97331, USA
*    Correspondence: stclais@piercecollege.edu

**Abstract:** In this study, we sought to investigate the impact of urbanization, the presence of concrete river bottoms, and nutrient pollution on microbial communities along the L.A. River. Six molecular markers were evaluated for the identification of bacteria, plants, fungi, fish, and invertebrates in 90 samples. PCA (principal components analysis) was used along with PAM (partitioning around medoids) clustering to reveal community structure, and an NB (negative binomial) model in DESeq2 was used for differential abundance analysis. PCA and factor analysis exposed the main axes of variation but were sensitive to outliers. The differential abundance of Proteobacteria was associated with soft-bottom sites, and there was an apparent balance in the abundance of bacteria responsible for nitrogen cycling. Nitrogen cycling was explained via ammonia-oxidizing archaea; the complete ammonia oxidizers, *Nitrospira* sp.; nitrate-reducing bacteria, *Marmoricola* sp.; and nitrogen-fixing bacteria *Devosia* sp., which were differentially abundant at soft-bottom sites (*p* adj < 0.002). In contrast, the differential abundance of several *cyanobacteria* and other anoxygenic phototrophs was associated with the impervious sites, which suggested the accumulation of excess nitrogen. The soft-bottom sites tended to be represented by a differential abundance of aerobes, whereas the concrete-associated species tended to be alkaliphilic, saliniphilic, calciphilic, sulfate dependent, and anaerobic. In the Glendale Narrows, downstream from multiple water reclamation plants, there was a differential abundance of cyanobacteria and algae; however, indicator species for low nutrient environments and ammonia-abundance were also present. There was a differential abundance of ascomycetes associated with Arroyo Seco and a differential abundance of *Scenedesmaceae* green algae and cyanobacteria in Maywood, as seen in the analysis that compared suburban with urban river communities. The proportion of Ascomycota to Basidiomycota within the L.A. River differed from the expected proportion based on published worldwide freshwater and river 18S data; the shift in community structure was most likely associated with the extremes of urbanization. This study indicates that extreme urbanization can result in the overrepresentation of cyanobacterial species that could cause reductions in water quality and safety.

**Keywords:** metabarcoding; microbial communities; nitrogen cycling; statistical modeling; water quality; urban river ecology

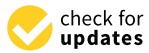



## 1. Introduction

The Los Angeles River has the potential to influence systems beyond its boundaries, such as estuarine environments at its outlet to the Pacific Ocean. In 2020, the County of Los Angeles' gross domestic product was USD 6.5 billion [1] and its population was

over 10 million [2]. Contamination, such as heavy metals, excess nutrients, coliform bacteria, and cyanide [3], have resulted from industrialization and high population. The L.A. River is a habitat for bacteria, fungi, fish, plants, and invertebrates that are sensitive to pollution. More recently, efforts have focused on protection and recognition of the river as a natural ecosystem, and part of that effort has been assessing the impacts of urbanization on the L.A. River ecosystems through eDNA sampling [4].

There have been few studies which have aimed to characterize the biome of the L.A. River; however, interest in characterizing microbial communities in this biome has increased in recent years [5,6]. The diversity of life, including fungi, bacteria, plants, fish, and invertebrates is indicative of ecosystem health. The presence or absence of certain "indicator" species reflect health and the presence of oxygen or degradation and pollution [7–10]. By investigating microbial community composition and identifying relative species abundance, ecosystem health can be compared among different locations subject to different pollutant profiles.

The L.A. River is unique and the impact of various types of urban pollution and infrastructure on microbial communities may be studied readily. The river runs through rural, suburban, and urban areas and the impact of population density can be assessed. The Los Angeles River was highly modified to facilitate flood control [10–12], due to flooding, which could be catastrophic.

A crucial question documented by Wenger et al. refers to the relationship between urbanization and the structure and function of microbial communities, which has not been well studied [13]. The question of how microbial communities may differ from one another in different land use areas and how urbanization may affect the proportions of different classes of microbes remains vital. The importance of this type of investigation was also underscored in the perspective of Antwis et al. on the most important areas of inquiry in microbial ecology [14]. In terms of urbanization, the modification or toxification of the environment may have influenced which microorganisms were present.

Habitats were diminished due to most of the L.A. River bottom being impervious concrete [15]. One of the key aspects of the paved condition is the decrease in plant life, which would absorb excess nitrogen in the environment through its roots. According to Wenger et al., an inquiry into the characteristics of piped or concrete paved tributaries as they influence biogeochemical processes represents one of the most important topics in urban stream ecology [13]. The presence of a concrete river bottom has been known to influence the oxygen content of freshwater, and this factor is expected to be one of the key factors which influences communities existing under a concrete-bottom condition. Nevertheless, if river organisms, such as oxygenic autotrophs, generate oxygen for the aboveground environment, it would help to offset such a concern as it performs a beneficial function. However, if cyanobacteria dominate, they may generate excess nitrogen which would not be absorbed by plants under paved conditions.

Since bacteria play a huge role in the breakdown of waste, nitrogen cycling, plant growth promotion, and pathogenicity, differences in bacteria warranted a closer look. A lack of oxygen in the underwater environment was expected to be one of the key factors which would influence communities under concrete-bottom conditions. Furthermore, concrete-paved rivers contribute to the urban heat island effect, which involves increased light intensity and heat [13]. Urban rivers generally have a cooling effect on a metropolis by virtue of the water that flows along them and the green spaces they support [16].

In the absence of rain, the L.A. River is fed by water from three water reclamation plants. Ackerman et al. found in 2003 that there were higher ratios of ammonia to nitrate near the water reclamation plants [10]. The benefits of using reclaimed water are obvious in terms of ecosystem services, as a river fed by recycled water would be expected to provide more habitat than a dry riverbed. The year-round supply of water has the potential to support wildlife and vegetation. The water sources have been shown to increase the $NO^{3-}$ concentration near the treatment plant sources, but it also would be expected to dilute the concentration of other pollutants, such as hydrocarbons from households and

industry pollutants, such as heavy metals. The proximity to a water reclamation plant could influence the diversity of bacterial sequences recovered from different sampling locations. A potential abundance of nitrate from water treatment plants was a concern historically at sites near Glendale [10]; however, the flow of water to wildlife would be expected to promote diversity. On balance, the river would be a dry ravine for most of the year due to the Mediterranean climate, if it were not for the releases from the water treatment plants.

In this study we sought to investigate the impact of urbanization, the presence of concrete river bottoms, and nutrient pollution on microbial communities along the L.A. River. This was achieved via meta-barcoding and the community analysis of environmental DNA (eDNA). Organisms that perform beneficial functions in the L.A. River ecosystem were identified and quantified from samples taken along the river [17]. This study focused on both eukaryotic and prokaryotic microbes, including archaea, bacteria, cyanobacteria, fungi, plants, and eukaryotic algae. Differences in the abundance of these organism types were measured and analyzed in order to test for statistically significant differences in composition between the sites of interest, i.e., differential abundance. This work contributes to a better understanding of the microbial ecology of the L.A. River ecosystem and helps identify urbanization impacts on microbial communities.

## 2. Materials and Methods

### 2.1. Sample Collection

The original data were generated as part of a BioBlitz program by University of California CALeDNA. CALeDNA is a collaboration of scientists creating a baseline of data for the biodiversity in California. Samples were collected by the UC CALeDNA team led by Miroslava Ramos, the project manager. Ninety replicated samples were collected from sediment over a 51-mile span of the channelized portion of the L.A. River and its tributaries. Three subsamples were taken from each sampling location and bulked after DNA extraction to capture a picture of the diversity within a 3-foot radius. In total, there were 180 subsamples.

Table 1 lists the sampling sites by their GPS coordinates for reference. The sampling sites were spread throughout the L.A. River Watershed. Tillman WRP is near the Sepulveda Dam. Note that Verdugo Wash flowed to Glendale Narrows, and Glendale also received water from the intermediary Glendale Water Reclamation Plant. Additionally, depicted is Arroyo Seco, a naturalized area that flows into the industrialized area of Maywood, providing contrast.

### 2.2. DNA Isolation and Amplification

DNA was extracted using the Qiagen DNEasy PowerSoil Kit. Six molecular markers specific to different kingdoms of life were amplified from the eDNA for amplicon sequencing. Amplicon libraries from each sample type with Illumina barcode adapters were sequenced on the MiSeq platform at 35,000 paired reads each. Quality control was performed in QIIME [18]. Cutadapt was used to remove Illumina adaptor sequences, and DADA2 was used for quality score trimming and the identification of unique ASVs. Taxonomies were assigned to amplicon sequence variants with an 80% likelihood cutoff from the CRUX database. A GreenGenes classifier was used. Each marker dataset was output into an ASV (amplicon sequence variant) table for downstream analysis using the Anacapa toolkit [19]. Table 1 shows the primer used for each marker in the dataset; in Table 2 metadata is provided for each of the samples.

**Table 1.** Tabulation of the types of genomic data that were available for the L.A. River [20].

| Marker | Description | Target Organisms | Forward Primer | Reverse Primer | Reference |
|---|---|---|---|---|---|
| FITS | Fungal rRNA Internal Transcribed Spacer | Fungi | GTCGGTAAAACTCGTGCCAGC | CATAGTGGGGTATCTAATCCCAGTTTG | Yang et al., 2018 [21] |
| 16S | Prokaryotic rRNA small subunit | Bacteria, archaea | GTGYCAGCMGCCGCGGTAA | GGACTACNVGGGTWTCTAAT | F: 515F and R: 806R, see Caporaso et al., 2012 [22] |
| 18S | Eukaryotic rRNA small subunit | Fungi, algae, protists | GTACACACCGCCCGTC | TGATCCTTCTGCAGGTTCACCTAC | Amaral-Zettler et al., 2009 [23]; Euk_1391f and EukBr |
| CO1 | Mitochondrial cytochrome oxidase subunit I | Animals | ATGCGATACTTGGTGTGAAT | GACGCTTCTCCAGACTACAAT | Gu et al., 2013 [24] |
| 12S | Mitochondrial rRNA small subunit | Fish, birds, snakes, insects | GGWACWGGWTGAACWGTWTAYCCYCC | TANACYTCnGGRTGNCCRAARAAYCA | Leray et al., 2013 [25] |
| PITS | Plant rRNA Internal Transcribed Spacer | Plants | GGAAGTAAAAGTCGTAACAAGG | CAAGAGATCCGTTGTTGAAAGTT | F: ITS5, White et al., 1990 [26]; R: 5.8S, Epp et al., 2012 [27] |

**Table 2.** The table of metadata for the L.A. River sites, showing the distribution of the samples across the site features.

| Sample No. | LA River Site | Latitude | Longitude | Habitat | River Condition |
|---|---|---|---|---|---|
| K0585_T9 | Arroyo Seco | 34.203154 | −118.166402 | Frequently submerged, intertidal, marsh | soft |
| K0593_C3 | Arroyo Seco | 34.203274 | −118.166417 | Terrestrial, not submerged | soft |
| K0594_E4 | Arroyo Seco | 34.202987 | −118.166335 | Terrestrial, not submerged | soft |
| K0595_B2 | Arroyo Seco | 34.203593 | −118.166448 | Terrestrial, not submerged | soft |
| K0595_L7 | Arroyo Seco | 34.203567 | −118.166415 | Terrestrial, not submerged | soft |
| K0595_T9 | Arroyo Seco | 34.204139 | −118.166314 | Terrestrial, not submerged | soft |
| K0597_M8 | Arroyo Seco | 34.20375 | −118.166481 | Terrestrial, not submerged | soft |
| K0599_L7 | Arroyo Seco | 34.20331 | −118.166408 | Frequently submerged, intertidal, marsh | soft |
| K0526_B2 | Bowtie Parcel | 34.108161 | −118.246186 | Fully submerged | soft |
| K0529_L7 | Bowtie Parcel | 34.108149 | −118.246176 | Fully submerged | soft |
| K0672_C3 | Bowtie Parcel | 34.108433 | −118.246959 | Fully submerged | soft |
| K0672_G5 | Bowtie Parcel | 34.108278 | −118.246926 | Fully submerged | soft |
| K0674_E4 | Bowtie Parcel | 34.108186 | −118.246584 | Fully submerged | soft |
| K0678_E4 | Bowtie Parcel | 34.108131 | −118.246003 | Fully submerged | soft |
| K0679_B2 | Bowtie Parcel | 34.108278 | −118.246341 | Fully submerged | soft |
| K0679_M8 | Bowtie Parcel | 34.108374 | −118.246774 | Fully submerged | soft |
| K0528_A1 | Bull Creek | 34.181558 | −118.497717 | Frequently submerged, intertidal, marsh | soft |
| K0528_E4 | Bull Creek | 34.182029 | −118.49771 | Frequently submerged, intertidal, marsh | soft |
| K0528_K6 | Bull Creek | 34.181975 | −118.497849 | Frequently submerged, intertidal, marsh | soft |
| K0529_K6 | Bull Creek | 34.181652 | −118.497718 | Frequently submerged, intertidal, marsh | soft |
| K0529_T9 | Bull Creek | 34.181651 | −118.497716 | Fully submerged | soft |

**Table 2.** *Cont.*

| Sample No. | LA River Site | Latitude | Longitude | Habitat | River Condition |
|---|---|---|---|---|---|
| K0530_A1 | Bull Creek | 34.181419 | −118.497763 | Frequently submerged, intertidal, marsh | soft |
| K0530_B2 | Bull Creek | 34.181342 | −118.497657 | Frequently submerged, intertidal, marsh | soft |
| K0530_E4 | Bull Creek | 34.1814 | −118.497865 | Frequently submerged, intertidal, marsh | soft |
| K0528_G5 | Compton Creek | 33.843656 | −118.206466 | Frequently submerged, intertidal, marsh | soft |
| K0528_L7 | Compton Creek | 33.843055 | −118.205667 | Fully submerged | soft |
| K0528_T9 | Compton Creek | 33.843328 | −118.2061 | Frequently submerged, intertidal, marsh | soft |
| K0529_A1 | Compton Creek | 33.843196 | −118.205854 | Frequently submerged, intertidal, marsh | soft |
| K0530_C3 | Compton Creek | 33.843311 | −118.206092 | Frequently submerged, intertidal, marsh | soft |
| K0530_K6 | Compton Creek | 33.842877 | −118.205544 | Frequently submerged, intertidal, marsh | soft |
| K0530_L7 | Compton Creek | 33.842749 | −118.205402 | Fully submerged | soft |
| K0530_M8 | Compton Creek | 33.843196 | −118.205854 | Frequently submerged, intertidal, marsh | soft |
| K0529_C3 | Elysian Valley | 34.083829 | −118.228152 | Fully submerged | concrete |
| K0672_T9 | Elysian Valley | 34.084621 | −118.228071 | Frequently submerged, intertidal, marsh | concrete |
| K0673_A1 | Elysian Valley | 34.084217 | −118.228066 | Frequently submerged, intertidal, marsh | concrete |
| K0673_G5 | Elysian Valley | 34.084227 | −118.228048 | Fully submerged | concrete |
| K0674_G5 | Elysian Valley | 34.08455 | −118.228053 | Fully submerged | concrete |
| K0676_B2 | Elysian Valley | 34.08449 | −118.228157 | Fully submerged | concrete |
| K0676_T9 | Elysian Valley | 34.084721 | −118.228145 | Fully submerged | concrete |
| K0677_A1 | Elysian Valley | 34.084482 | −118.228157 | Frequently submerged, intertidal, marsh | concrete |
| K0593_T9 | Glendale | 34.155282 | −118.275211 | Fully submerged | concrete |
| K0594_L7 | Glendale | 34.15459 | −118.276618 | Fully submerged | concrete |
| K0596_C3 | Glendale | 34.155107 | −118.275459 | Fully submerged | concrete |
| K0596_E4 | Glendale | 34.154774 | −118.27637 | Frequently submerged, intertidal, mars | concrete |
| K0596_L7 | Glendale | 34.154918 | −118.276231 | Fully submerged | concrete |
| K0596_T9 | Glendale | 34.154973 | −118.275799 | Fully submerged | concrete |
| K0597_K6 | Glendale | 34.154997 | −118.275944 | Fully submerged | concrete |
| K0597_L7 | Glendale | 34.155157 | −118.27542 | Fully submerged | concrete |
| K0526_C3 | Glendale Narrows | 34.102813 | −118.242742 | Fully submerged | concrete |
| K0526_G5 | Glendale Narrows | 34.103427 | −118.242642 | Fully submerged | concrete |
| K0529_B2 | Glendale Narrows | 34.103109 | −118.242634 | Fully submerged | soft |
| K0529_G5 | Glendale Narrows | 34.103652 | −118.242686 | Fully submerged | concrete |

**Table 2.** *Cont.*

| Sample No. | LA River Site | Latitude | Longitude | Habitat | River Condition |
|---|---|---|---|---|---|
| K0529_M8 | Glendale Narrows | 34.103251 | −118.242645 | Fully submerged | concrete |
| K0672_B2 | Glendale Narrows | 34.10274 | −118.242669 | Fully submerged | concrete |
| K0678_B2 | Glendale Narrows | 34.103274 | −118.242544 | Fully submerged | concrete |
| K0678_K6 | Glendale Narrows | 34.103437 | −118.24275 | Fully submerged | concrete |
| K0672_A1 | Long Beach | 33.762909 | −118.202355 | Fully submerged | soft |
| K0674_M8 | Long Beach | 33.762738 | −118.202271 | Fully submerged | concrete |
| K0676_M8 | Long Beach | 33.762683 | −118.202126 | Fully submerged | concrete |
| K0677_B2 | Long Beach | 33.762833 | −118.202418 | Fully submerged | concrete |
| K0677_E4 | Long Beach | 33.762907 | −118.202298 | Fully submerged | concrete |
| K0677_L7 | Long Beach | 33.762841 | −118.20235 | Fully submerged | concrete |
| K0678_L7 | Long Beach | 33.762906 | −118.202305 | Fully submerged | soft |
| K0701_C3 | Long Beach | 33.76269 | −118.202303 | Fully submerged | concrete |
| K0527_A1 | Maywood | 33.986755 | −118.171412 | Frequently submerged, intertidal, marsh | concrete |
| K0527_C3 | Maywood | 33.988033 | −118.172607 | Fully submerged | concrete |
| K0527_E4 | Maywood | 33.987023 | −118.171842 | Fully submerged | concrete |
| K0527_K6 | Maywood | 33.986686 | −118.171342 | Fully submerged | concrete |
| K0527_L7 | Maywood | 33.987668 | −118.172288 | Fully submerged | concrete |
| K0527_T9 | Maywood | 33.986617 | −118.171324 | Fully submerged | concrete |
| K0539_L7 | Maywood | 33.986776 | −118.17165 | Fully submerged | concrete |
| K0593_G5 | Sepulveda Dam | 34.168961 | −118.475292 | Fully submerged | soft |
| K0594_A1 | Sepulveda Dam | 34.168698 | −118.475195 | Fully submerged | soft |
| K0594_T9 | Sepulveda Dam | 34.168961 | −118.475292 | Fully submerged | soft |
| K0595_G5 | Sepulveda Dam | 34.168941 | −118.47461 | Terrestrial, not submerged | soft |
| K0597_T9 | Sepulveda Dam | 34.1688 | −118.475049 | Fully submerged | soft |
| K0599_G5 | Sepulveda Dam | 34.16868 | −118.474846 | Frequently submerged, intertidal, marsh | soft |
| K0599_K6 | Sepulveda Dam | 34.168906 | −118.475125 | Fully submerged | soft |
| K0599_T9 | Sepulveda Dam | 34.168758 | −118.474733 | Rarely submerged, wetland, arroyo | soft |
| K0593_A1 | Tujunga Wash | 34.258032 | −118.386781 | Fully submerged | concrete |
| K0593_E4 | Tujunga Wash | 34.258403 | −118.386614 | Fully submerged | concrete |
| K0595_M8 | Tujunga Wash | 34.257481 | −118.386845 | Fully submerged | concrete |
| K0596_B2 | Tujunga Wash | 34.258667 | −118.386473 | Fully submerged | concrete |
| K0597_E4 | Tujunga Wash | 34.258716 | −118.386376 | Fully submerged | concrete |
| K0599_A1 | Tujunga Wash | 34.258424 | −118.386387 | Fully submerged | concrete |
| K0599_E4 | Tujunga Wash | 34.258395 | −118.386592 | Fully submerged | concrete |
| K0599_M8 | Tujunga Wash | 34.258016 | −118.386744 | Fully submerged | concrete |
| K0593_L7 | Verdugo Wash | 34.203216 | −118.237654 | Fully submerged | soft |
| K0595_A1 | Verdugo Wash | 34.202985 | −118.237755 | Fully submerged | soft |
| K0596_G5 | Verdugo Wash | 34.202611 | −118.237615 | Fully submerged | soft |

For this differential abundance analysis, computation focused on the bacteria and fungi. However, the results of the differential abundance analysis may also include algae and nematodes, for example. Table 3 shows the covariates that were contrasted in DESeq2.

**Table 3.** List of covariates that were tested for association with a differential abundance of bacterial and fungal taxa.

| Marker | Covariate | Factor Levels Tested |
|--------|-----------|----------------------|
| 16S | LA River Site | Glendale Narrows, Verdugo Wash |
| 16S | River Condition | Soft-Bottom, Concrete |
| 16S | Habitat | Frequently Submerged, Fully Submerged |
| FITS | Habitat | Frequently Submerged, Fully Submerged |
| FITS | LA River Site | Maywood, Arroyo Seco |

### 2.3. Statistical Approach

The goal of this project was to examine sample diversity using a variety of methods using a Euclidean distance matrix [28]. The Euclidean distance is given by [29]:

$$d(j_1, j_2) = [(X1j_1 - X1j_2)^2 + \cdots + (Xnj_1 - Xnj_2)^2]^{1/2}$$

The methods utilizing the Euclidean dissimilarity measure will include the neighbor joining of samples [30], the UPGMA of samples [30], heatmap visualization using the chi-square standardization of samples, and PAM (partitioning around medoids) clustering applied to PCA. Ranacapa [31] was used to perform a PERMANOVA beta diversity test and visualize with principal coordinates analysis (PCoA) to help with hypothesis development.

PAM clustering was applied to PCA to investigate whether samples cluster by location in an unsupervised model and whether the PCA reflected a spatial relationship inherent in the genetic distances. The PAM function from the cluster package was used [32]: First, $K$ representative medoids are arbitrarily selected, then swapping cost $C_{ih}$ to swap medoid $h$ and non-medoid $i$ is calculated. If the resulting value is negative, then the medoid and non-medoid are swapped. The process is repeated until there is no change. Principal components analysis reveals population stratification and PAM is used for classification of samples.

The classification of samples was expected based on the taxonomic composition of samples; that is, if there were differentially abundant taxa between groupings then separation into different PAM clusters would be expected. To select the optimal number of clusters $K$, the PAM model with the highest average silhouette value was selected. The factor analysis of the most important taxon features in the PCA for each marker dataset gave some preliminary evidence about which particular taxa may be differentially abundant. Relative abundance was compared for important plant taxa using a pivot table in Microsoft Excel.

### 2.4. Chi Square Test of Proportions for the 18S Marker

The data were published originally as "Table 2, Richness of Main Taxonomic Groups of Fungi in Freshwater Ecosystems" from a study that has count data for the main taxonomic groups of fungi in freshwater ecosystems that can be used as a comparison [33]. The information captures data from 22 publicly available datasets from around the world. The initial exploration of the data revealed that there were few Cryptomycota and Chytridiomycota identified in the pooled L.A. River samples. The chi-square test tested whether the proportion of Ascomycota:Basidiomycota in the L.A. River differed significantly from that of freshwater and river environments in the published data. The hypotheses that were tested for this analysis are contained in Supplemental Materials.

Overdispersion is common in taxonomic count data for environmental samples. The model that was implemented in DESeq2 to answer these research questions was a negative

binomial model. In these data, zero inflation is also suspected. The way that DESeq2 dealt with overinflation in this analysis was to analyze only positive counts. Exploratory plots for dispersion in the fungi dataset were generated to further investigate the appropriateness of the model (see Supplemental Materials).

## 2.5. Differential Abundance Analysis

For differential abundance analysis, DESeq2 was employed [34]. The DESeq2 package has handled RNA-seq or ChIP-seq, metabarcoding ASV tables, and any similar genomic data that consisted of counts. The goal was to correct some problems associated with using chi-square test and the Poisson distribution for these types of data, which may not effectively control a Type I error [34].

It was assumed that the number of reads in sample $j$ assigned to gene or taxon $i = Kij \sim NB(\mu ij, \sigma^2)$ follows a negative binomial distribution (NB), which is commonly used for the modelling of data in the presence of overdispersion [34].

The following further assumptions were made:

1. The mean parameter is the expectation value for $Kij$ and is proportional to the actual number of sequence counts for gene $i$ under the experimental condition $\rho$. The size factor is also accounted for, which is essentially the coverage or sequencing depth of the genetic library for each sample.
2. The variance $\sigma^2$ is the sum of the shot noise and the raw variance.
3. The model uses a pooled variance from genes (or taxa) with similar count values to estimate the per gene raw variance.

$Kij$ follows a Poisson distribution. If the rate that fragments are assigned to known sequences depends on a random variable $Rij = rij$, and the size factor, $sij$, then when $Rij$ is modeled by the gamma distribution, $Kij \sim NB(\mu ij, \sigma^2)$, the cycle has been completed.

In terms of fitting the model, data exist in a $n \times m$ table of $Kij$ counts: $i = 1 \ldots n$ genes in $j = 1 \ldots m$ samples. The parameters used were:

1. $m$ size factors, including 1 for each sample.
2. $n$ expression strength parameters $qip$ for each condition $\rho$. In other words, the expectation values for the abundance of counts for gene or taxon $i$ are proportional to $qip$.
3. The pooled variance parameter simulates the dependence of $Vip$ on the expectation value for the mean, $qip$, for each condition $\rho$.

The size factor $sij$ allows comparisons between samples with different sequencing depths. Size factors are estimated via the median of observed count ratios [34]. $qip$ is estimated through a transformation of the average counts from $j$ samples under condition $\rho$. The fit can be applied to small numbers of replicates using local regression to estimate the raw variance. The method is a gamma family GLM for a local regression that implements R locfit.

A hypothesis rejection in DESeq would mean that the difference in counts between two samples was larger than would be expected if the samples were replicates from the same individual or tissue [34]; the rejection does not indicate what is responsible for the difference. A rejection shows that a taxon, protein, or gene count was differentially abundant between two samples. However, a hypothesis rejection would not reveal if it was more different than what would typically be seen if two separate locations along the same river were sampled. It would also not reveal whether the difference would have a greater magnitude than if one compared the differential abundance of that taxa between two different rivers. It empowers the user to detect differences, while controlling the Type I error. Volcano plots were subsequently visualized in SystemPipeR [35] and Enhanced Volcano [36].

## 3. Results

The Unweighted Unifrac distance method coupled with PERMANOVA, visualized by PrinCoA, was the most sensitive for the detection of differences between groups based on sampling site, habitat, or depth. The chi-squared standardized heatmap was not sensitive.

PCA alone was not sensitive, although the factor loadings were useful for revealing the few important taxa that differed between samples. PAM coupled with PCA was more useful for identifying highly similar groups of samples and elucidating community structure. PCA with PAM gave a better visualization than the hierarchical clustering methods for this sample size, although overall, the PAM and UPGMA results were very similar.

Table 3 shows the medians and ranges for taxon abundance and sequences per sample. The FITS marker had a median number of sequences per sample of 18,157. Table S1 displays the summary statistics resulting from the NJ (neighbor joining) and UPGMA (unweighted pair group method with arithmetic mean) tree analyses in R phyloseq. As shown in Table S1, the branch length means were similar, but the variance is higher for neighbor joining, with respect to the FITS marker. A higher variance for neighbor joining would be expected.

Depicted in Figure S3 is the PCA for the fungal ITS sequences that were recovered from the L.A. River sediment samples. The first two principal components capture about 37% of the variation in the data. Fungi samples separate high on PC 2 based on the abundance of *Penicillium*, which may be important for the decomposition of leaf litter along the river, and *Cladosporium* sequences, which produce the antibiotic and antimalarial metabolite Cladosporin [37]. Low on PC 2, the separation is based on the abundances of the *Desmodesmus armatus* and *Desmodesmus* sp. variants of algae, especially in Maywood, Glendale Narrows, Glendale, and Elysian Valley. These genera have been known to break down radioactive materials.

Other results from the DESeq2 analysis, showed that in frequently submerged river condition samples, there was a significantly higher abundance of fungi and less bacteria, when compared with submerged samples. The volcano plot showing the large number of significant results for fungi based on the FITS marker is visualized in Figure 1. In frequently submerged sediment samples, *Capniodales* sp. were differentially abundant based on the adjusted *p*-values ($p < 1 \times 10^{-13}$), as well as *Penicillium* sp. ($p < 0.0005$). Notably, *Tricladium angulatum* ($p < 1.5 \times 10^{-46}$), *Monocillium tenue* ($p < 2.5 \times 10^{-39}$), *Acremonium nepalense* ($p < 5 \times 10^{-30}$), and *Peziza badia* ($p = 9.5 \times 10^{-15}$) were also significantly more abundant in frequently submerged samples.

As shown in Table 3, the 16S assay had a strong median number of sequences per sample at 15,178. This shows that the sample had a good sequencing depth. As shown in Table 4, the branch length means are similar but the variance is about 50,000 units higher for neighbor joining, with respect to the 16S marker. The rooted and unrooted trees both indicated k = 5 for the number of clusters in the community of bacteria.

**Table 4.** Summary statistics from the neighbor joining and UPGMA trees for each marker. The trees were generated from the Euclidean distance matrix. The tree topological distances have been provided in the far-right column.

| L.A. RIVER | Branch Length NJ | | Branch Length UPGMA | | NJ vs. UPGMA |
|---|---|---|---|---|---|
| **Marker** | **Mean** | **Variance** | **Mean** | **Variance** | **Tree Distance** |
| **FITS** | 1657 | 5,419,114 | 1585 | 4,124,851 | 8195 |
| **16S** | 620 | 460,349 | 609 | 417,224 | 2473 |
| **18S** | 2018 | 5,534,355 | 1978 | 4,278,736 | 10,919 |
| **COI** | 2312 | 8,746,132 | 2114 | 6,010,691 | 9697 |
| **12S** | 634 | 4,710,694 | 1585 | 4,124,851 | 12,130 |
| *PITS* | 1457 | 6,728,373 | 1351 | 4,241,554 | 8516 |

Figure S5 shows the PCA for the bacterial 16S DNA sequences that were recovered from the L.A. River sediment samples. The first two principal components capture around 42% of the variation in the data. Bacteria DNA samples are separated by numerous

important taxa factor loadings, such as the abundance of Erythrobacteracea, Proteobacteria, and *Oscillatoriales cyanboacterium*.

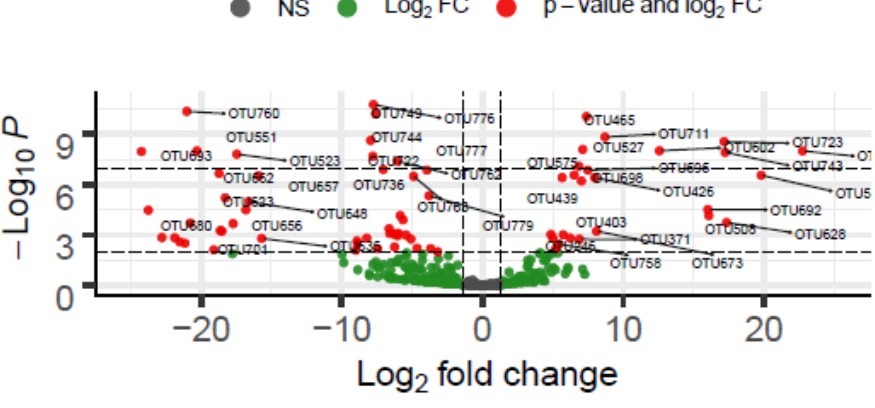

**Figure 1.** The volcano plot demonstrated the large number of taxa that were differentially abundant between Maywood and Arroyo Seco, with regard to fungi. The differential abundance analysis in DESeq2 for the FITS marker yielded a large number of interesting fungi associated with one location or another (101 significant taxa were detected). NS = not significant, FC = fold change. The OTUs with negative log fold change values were more abundant in Maywood; the OTUs with positive log fold change values were more abundant in Arroyo Seco.

Among others, samples from Maywood and Glendale scored low on PC 2 in terms of high cyanobacteria abundance. Figure 2 shows that the PCA plot for the 16S samples was color coded, corresponding to the best PAM clustering. The best PAM clustering in this case was k = 4 with the highest average silhouette width. The samples in the second cluster, colored red, are from Glendale Narrows. The third cluster, colored green, is mostly made up of sediment samples from Maywood and Glendale. The blue and black clusters are made up of a mixture of the remaining sites.

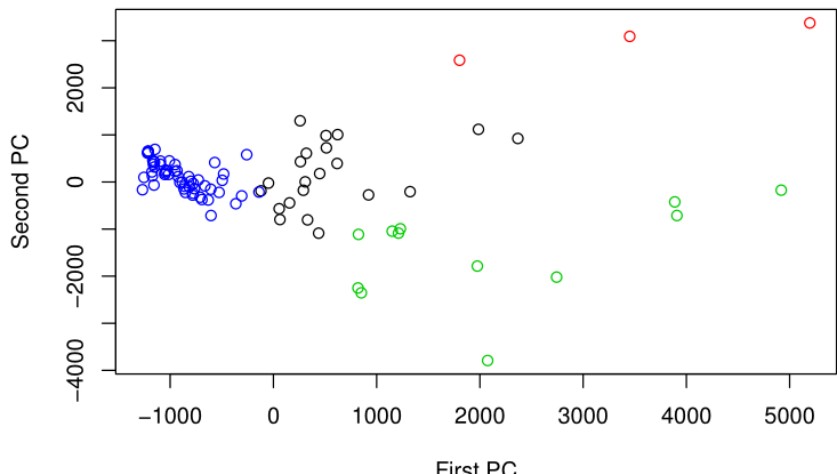

**Figure 2.** PCA for bacterial identified sequences from the 16S marker by sample, color coded by the best PAM clustering. Each point represents a sample. Note that there is evidence of overdispersion, in particular, high on PC1. The points with the same color were classified by PAM as belonging to the same cluster.

Among the bacteria with a differentially higher abundance of 16S sequences in Glendale Narrows, *Cyanobacteria microcystis* ($p < 1.5 \times 10^{-7}$) and *Oscillatoriales cyanobacterium* ($p < 3 \times 10^{-14}$). *Verrucomicrobia* were also differentially more abundant in Glendale Narrows ($p < 4 \times 10^{-23}$). On the other hand, the alphaproteobacteria *Devosia* from Rhizobiales had differentially higher counts of sequences in samples from Verdugo Wash. These clusters helped inform the DESeq Analysis for Glendale vs. Verdugo Wash and soft-bottom vs. concrete contrasts, the results of which are shown in Tables 5 and 6.

**Table 5.** The results of the differential abundance analysis for Glendale vs. Verdugo Wash. Positive log fold change results represent sequences that were differentially abundant at the Glendale site. Negative log fold changes represent sequences that were differentially abundant at the Verdugo Wash site.

| Taxon | Log$_2$ Fold Change | *p*-adj | Ecological or Metabolic Function and Pathogenicity |
|---|---|---|---|
| *Prosthecobacter* sp. | 22.09927 | $3.71 \times 10^{-23}$ | possible pathogen, anaerobic, tubulin-like genes, low nutrient environments |
| *Dechloromonas* sp. | 34.31956 | $1.53 \times 10^{-41}$ | may oxidize benzene |
| *Devosia* sp. | −22.258 | $5.73 \times 10^{-5}$ | nitrogen fixer |
| *Bacillus* sp. | −25.3115 | $1.67 \times 10^{-5}$ | many beneficial species |
| Chromatiaceae (unclassified) | 23.78784 | $1.22 \times 10^{-6}$ | purple sulfur bacteria, use sulfide to fix carbon and generate oxygen |
| *Sandaracinobacter* sp. | −30.519 | 0.009416 | metabolism of sulfide to cysteine (or from serine) |
| *Chloroflexaceae* (unclassified) | 25.68591 | 0.000938 | green non-sulfur bacteria, many heat-loving anoxygenic photoheterotrophs [38,39] |
| endosymbiont of *Ridgeia piscesae* | −22.3636 | 0.00014 | gammaproteobacterium, symbiont of a tubeworm |
| anaerobic bacterium MO-CFX2 *Chloroflexi* | −6.85917 | $4.08 \times 10^{-6}$ | |
| Rhodocyclales (unclassified) | 17.1087 | $4.15 \times 10^{-8}$ | nitrogen fixing or nitrogen reducing |
| *Phormidium setchellianum* | 33.82601 | $2.58 \times 10^{-14}$ | potential cause of gastroenteritis, concentrates caused neuro- and hepato-toxicity in mice [40] |
| *Cytophaga xylanolytica* | 20.18264 | 0.000268 | xylan degrading, does well in sulfogenic and methanogenic environments, anaerobic and gliding |
| *Synechococcus* sp. | −23.4117 | 0.002659 | photolysis of sulfide or water, produces neurotoxins [41] |
| *Scenedesmaceae* (unclassified) | 11.0032 | 0.000123 | green algae, may degrade radioactive materials |
| *Flavobacterium* sp. | 8.245038 | 0.000199 | often associated with plant resistance to pathogens |
| *Oscillatoriales* cyanobacterium HF1 | 7.271474 | 0.005122 | cyanobacterium which may cause illness or death in humans and animals |
| *Tetradesmus obliquus* | 10.11933 | 0.001645 | produces valuable saturated and unsaturated esters, extract has anticancer and antimicrobial effects [42,43] |
| *Microcystis* sp. | 28.7773 | $1.03 \times 10^{-7}$ | cyanobacterium which is toxic to humans [44] |
| *Rhodocyclaceae* bacterium enrichment culture clone Y62 | 28.91261 | $5.24 \times 10^{-5}$ | nitrogen fixing or nitrogen reducing |



**Table 6.** Positive log fold change results represent sequences that were differentially abundant at the soft-bottom sites. Negative log fold changes represent sequences that were differentially abundant at the concrete sites.

| Taxon | Log$_2$ Fold Change | *p*-adj | Ecological or Metabolic Function and Pathogenicity |
|---|---|---|---|
| *Oscillatoriales cyanobacterium* YACCYB599 | −25.207183 | $3.06 \times 10^{-23}$ | cyanobacteria, which may cause illness or death in humans and animals |
| *Chroococcus subviolaceus* | −24.66764915 | $4.55 \times 10^{-23}$ | freshwater or high salinity environments, cyanobacteria which can survive with low $O_2$ [45] |
| *Haliea* sp. | −24.50212313 | $4.55 \times 10^{-23}$ | marine gamma proteobacterium, which tolerates up to 12% salinity [46,47] |
| *Halomonas* sp. | 24.49667323 | $3.81 \times 10^{-31}$ | chloride and saline tolerance |
| *Marmoricola* sp. | 24.12963073 | $1.43 \times 10^{-27}$ | denitrifying bacteria [48] |
| Alpha proteobacterium LS7-MT | 10.00393321 | $8.21 \times 10^{-09}$ | methanol oxidizer, lives in high temperatures [49] |
| *Nitrosarchaeum koreense* | 9.188395232 | $2.37 \times 10^{-18}$ | aerobic ammonia-oxidizing archaea [50] |
| Microcystaceae (unclassified) | −8.382519826 | 0.001244 | common eutrophic bloomer, toxin-producing cyanobacterium |
| *Acidobacterium* sp. SCGC AAA007-P13 | 7.849119335 | $3.12 \times 10^{-7}$ | potential saprobe |
| *Oscillatoriales cyanobacterium* IRH12 | −7.732408042 | $4.32 \times 10^{-8}$ | cyanobacterium, which may cause illness or death in humans and animals |
| *Roseisolibacter agri* | −7.389766623 | 0.000539 | grows in low oxygen environments [51] |
| *Pleurocapsa concharum* | −7.310779292 | $1.03 \times 10^{-7}$ | ostracod-dependent cyanobacterium [52] |
| *Devosia* sp. | 7.242636088 | $5.51 \times 10^{-7}$ | nitrogen-fixing bacteria |
| *Nitrospira* sp. enrichment culture clone LD3 | 6.970043209 | 0.001616 | nitrifying bacteria, nitrite-oxidizing bacteria |
| Gamma proteobacterium SCGC AAA007-P21 | 6.533527317 | $1.83 \times 10^{-13}$ | uncultivated bacterioplankton |
| alpha proteobacterium Schreyahn_AOB_Aster_Kultur_5 | 6.503508981 | 0.001529 | cultured alphaproteobacterium |
| Chlamydomonadales (unclassified) | −6.479686479 | 0.000178 | green algae [53] |
| *Chloronema giganteum* | −6.382235759 | 0.000425 | photoautotrophic, anoxygenic green non-sulfur bacteria [54] |
| *Chamaesiphon* sp. | −6.230017507 | 0.002384 | widely distributed cyanobacterium [55] |
| *Altererythrobacter* sp. | 6.02052523 | 0.007591 | alkaline or salt tolerant aerobic phototroph, anoxygenic [56–58] |
| Mycobacteriaceae (unclassified) | 5.990283542 | 0.000524 | potential human and animal pathogens |
| Acidobacteriaceae (unclassified) | 5.737312813 | $2.78 \times 10^{-6}$ | likely saprobe of plant organic matter |
| *Candidatus Viridilinea mediisalina* | −5.72085055 | 0.009826 | anaerobic phototroph, salt-tolerant and prefers alkaline environments [59] |
| Veillonellaceae bacterium 6–15 | −5.56037325 | $2.59 \times 10^{-5}$ | bacterial vaginosis |
| *Phormidium setchellianum* | −5.548460876 | 0.000699 | cyanobacterium with possible antitumor agents, neuro and hepatotoxicity |

**Table 6.** *Cont.*

| Taxon | Log$_2$ Fold Change | *p*-adj | Ecological or Metabolic Function and Pathogenicity |
|---|---|---|---|
| *Calothrix* sp. UAM 374 | −5.531306605 | 0.003193 | cyanobacterium, which grows on plants and hard substrates [60] |
| Candidatus *Nitrosocosmicus* sp. | 5.344610141 | 0.0001 | aerobic ammonia-oxidizing archaea |
| *Treponema stenostreptum* | −5.019693824 | 0.003193 | syphilis relative |
| Leptolyngbyaceae (unclassified) | −4.952937198 | 0.001067 | thermophilic and potentially iron-loving cyanobacterium [61] |
| Holophagaceae (unclassified) | −4.934291389 | 0.000964 | anaerobic dweller of freshwater sediments [62] |
| Xanthomonadaceae bacterium | −4.711954167 | 0.002384 | potential phytopathogens |
| *Leptolyngbya geysericola* | −4.711366069 | 0.005914 | alkaline tolerant non-heteroctic cyanobacterium, produces calcite on microplastics [63] |
| Caldilineales bacterium | 4.50039412 | $4.71 \times 10^{-6}$ | thermophilic and anaerobic [64] |
| *Fusibacter* sp. enrichment culture | −4.35065315 | 0.009823 | thiosulfate reducing, potentially halotolerant |
| *Desulfomicrobium* sp. | −4.16646108 | 0.002439 | oxidizes sulfide and arsenate in the presence of $CO_2$ and acetate [65], reduces nitrate to ammonium [66] |
| Oscillochloridaceae (unclassified) | −3.874861377 | 0.005914 | anoxygenic phototrophic bacteria [38,67] |
| Pleurocapsales (unclassified) | −3.695598612 | 0.009826 | cyanobacterium from calcareous environments |
| *Vicinamibacter silvestris* | 3.602101991 | 0.002384 | polyphosphate accumulating organisms |
| Firmicutes (unclassified) | 2.378738101 | 0.004923 | high abundance in suburban rivers, negatively correlated with ammonia concentration |
| *Stenotrophobacter terrae* | 2.253024076 | 0.008829 | opportunistic pathogen |
| Vicinamibacteraceae (unclassified) | 2.126473277 | 0.00044 | degrades chitin [68] |
| Actinobacteria (unclassified) | 2.033767588 | 0.003193 | many denitrifying bacteria [69,70] |

The soft-bottom river condition was associated with a differentially higher abundance of Alphaproteobacteria and a decreased abundance of *Cyanobacteria pleurocaps* ($p < 1 \times 10^{-6}$) and *Phormidium* ($p < 0.0007$), *Oscillatoria* ($p < 3 \times 10^{-23}$), and *Chroococci* ($p < 5 \times 10^{-23}$) when contrasted with concrete sites. Notably, *Devosia* was more abundant in soft bottoms ($p < 6 \times 10^{-7}$), whereas *Desulfomicrobium* ($p < 0.003$) was more abundant under concrete-bottom conditions. On the other hand, Verrucomicrobia and Haliaceae family Proteobacteria were differentially abundant under soft-bottom conditions ($p < 5 \times 10^{-23}$, $p = 0.01$, respectively).

Most of the bacteria that were differentially expressed in the concrete sites were cyanobacteria and autotrophs. There was also a trend toward a differentially high abundance of DNA sequences from potential human and plant pathogens, including the potential plant pathogen *Xanthomonas*, Clostridia, and bacteria related to the agents that cause reproductive infections. Nevertheless, the soft-bottom sites also had differentially high abundances of Norcardiaceae and Verrucomicrobia, which are also potential pathogens. For the concrete sites, there was a less clear picture of the nitrogen cycle when considering the bacteria alone. There was a clear picture of the nitrogen cycle for the soft-bottom sites, as well as a candidate species for phosphate accumulation.

The highest number of assigned sequences per sample was for the 18S marker, as shown in Table 2. This suggests that the highest overall sequencing depth was for the 18S assay. As shown in Table 4, the branch length means were both near 2000 but the variance was around 125,000 units higher for neighbor joining, with respect to the 18S marker. For both tree topologies, k = 4 is apparent for the number of clusters in terms of 18S sequences identified by the assay.

In Figure S6, the PCA for the 18S DNA sequences that were recovered from the L.A. River sediment samples is shown. The first two principal components capture around 46% of the variation in the data. The PCA by sample for 18S validates the FITS results, because the samples scored low on PC 2 based on factor loadings for *Desmodesmus* and other *Scenedesmaceae* taxa of algae. Further, samples scored high on PC 2 based on the *Podocopida* and *Cypridida* high relative sequence abundance. *Podocopida* is a crustacean that comprises freshwater and brine-dwelling groups [71]. The *Cyprididae* are a group of freshwater Ostracods [72]. Figure S7 shows the 18S PCA color coded by the best PAM clustering, which was k = 5, with the highest average silhouette width. The red samples in cluster 2 were all from Glendale. Cluster 5, in light blue, corresponds to the Long Beach sediment samples. Considering the spatial heterogeneity displayed by the samples, there is a sense that the genetic material is funneling into Long Beach, reflecting the physical landscape. The fourth cluster, in dark blue, is composed of Sepulveda Dam, Tujunga Wash, and Arroyo Seco.

The observed alpha diversity for fungi sequences based on the 18S marker is shown in Figure 3. Los Angeles River proportions of Ascomycota and Basidiomycota were compared to freshwater and river habitats worldwide. The equality of these proportions were tested on a chi-square distribution. The results showed that the proportions of Ascomycota and Basidiomycota in the L.A. River differed significantly from freshwater and river environments worldwide, based on published 18S data [33].

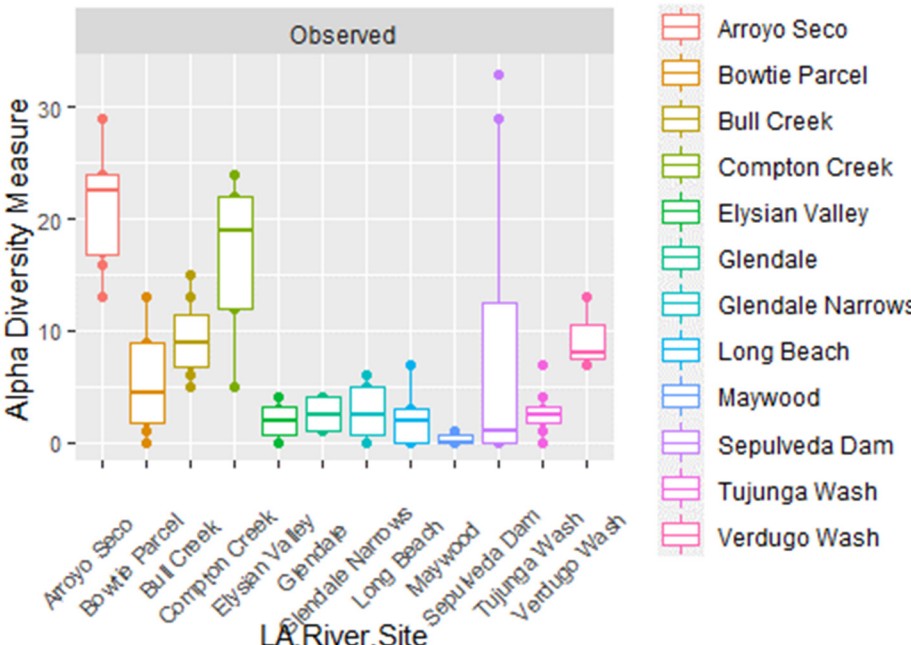

**Figure 3.** The boxplot of observed alpha diversity shows that the species richness for Ascomycota is the highest in Arroyo Seco, Bull Creek, Compton Creek, and Maywood.

The data that were used for this part of the analysis are publicly available [33] as amplicon sequence variants tables, also known as ASVs or OTUs. OTU stands for operational taxonomic unit. Essentially, these tables have the counts of sequences that were identified from organisms in the environment. The goal is to compare the proportions of different divisions of fungi in the L.A. River to other environments.

In Figure 4, the mosaic plot for the chi-square test of proportions for river habitats worldwide versus those of the L.A. River is shown. The values for the Ascomycota and Basidiomycota in the L.A. River and in worldwide river habitats display a gap between them. This shows that these proportions differ significantly from what one expect if they belonged to the same population. The results of the chi-square test for the equality of proportions shows that the values of Ascomycota and Basidiomycota for the L.A. River are not equal to the proportions of Ascomycota and Basidiomycota in freshwater habitats ($p < 0.0005$) or river habitats ($p < 1 \times 10^{-11}$) described in Lepère's analysis of worldwide freshwater data. In terms of the river habitats, the proportion of Ascomycota to Basidiomycota is 21.5–39.2% higher in the L.A. River. Furthermore, for the freshwater habitat comparisons, the proportion of Ascomycota to Basidiomycota is between 7.3–25.74% higher for the L.A. River, based on the 95% confidence intervals. When comparing the mosaic plots in Figure S9 and Figure 4, the gap between the values of Ascomycota and those of Basidiomycota appear smaller for the L.A. River compared to freshwater habitats in the study by Lepère et al. [33], compared with river environments.

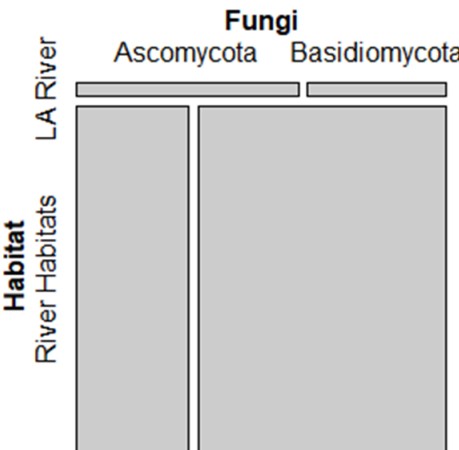

**Figure 4.** The mosaic plot shows that there is a difference in the proportion of Ascomycetes to Basidiomycetes in the L.A. River compared to river habitats worldwide [33]. This gap was larger than the gap shown in Figure S9 for freshwater habitats.

The alpha diversity analysis for Ascomycetes is plotted in Figure 3. The mosaic plot shows that the sites that had the most Ascomycota species were detected at Arroyo Seco, Bull Creek, Compton Creek, and Maywood. Maywood had much variability: two points were outliers with high counts >25, whereas most values were near zero. It is also interesting to note that more than 50 taxa of Ascomycota were identified only to the family level, and some of these may represent heretofore uncharacterized Ascomycetes. Based on these results, an interesting junction of the L.A. River to investigate Ascomycete sequences to a deeper level would be Arroyo Seco and Maywood, which were geographically connected.

The plot of alpha diversity for all fungi, given in Figure 3, shows which sites had the most different types of fungi in any division. Overall, there were 132 taxa of fungi identified. Arroyo Seco, Bull Creek, Compton Creek, Maywood, and Verdugo Wash accumulated the most taxa. An interesting aspect regarding this point is that out of the 132 taxa of fungi, over 30% were Ascomycetes identified only to the family level.

The COI marker performed well in terms of median sequences per sample, which was 18,555. As shown in Table 3, the branch length mean is about 200 units longer for NJ and the variance is about 275,000 units higher for neighbor joining, with respect to the COI marker. For both tree topologies, k = 3 is apparent for the number of clusters in terms of COI sequences identified. This seems to reflect that the animal diversity detected by the assay has less breadth than the biodiversity captured by 16S or FITS in this instance.

In Figure S10, the PCA for the COI DNA sequences that were recovered from the L.A. River sediment samples is shown. The first two principal components capture about 33% of the variation in the data. The COI assay captured a picture of lower diversity for the sequences. Samples score low on PC 2 based on the relative abundance of the *Dicrotendipes* species, i.e., non-biting bloodworms [73]. Additionally, low on PC 2 were samples with a high relative abundance of *Eucypris virens*, a cyprididine ostracod [74]. The presence of bloodworms is an indicator that other animals are present in the River and is a positive indicator of ecosystem health. The ostracod *E. virens* is sensitive to heavy metal pollution; therefore, the presence of this ostracod in significant numbers is also an indicator of ecosystem health.

The PCA plot for the COI samples color coded by the best PAM clustering is shown in Figure S11. The best PAM clustering in this case was k = 3, which exhibited the highest average silhouette width. For the COI sequences, 73 of the samples fall into the first cluster shown in black, ranging from Bowtie Parcel to Verdugo Wash. The second cluster, in red, is composed of Glendale and Sepulveda sediment samples. The third cluster, shown in green, is made up of only two samples from Tujunga Wash and Glendale. This supports the observation that samples were similar to this marker.

The abundance of sequences per taxon for 12S was lower than the other markers assayed at a maximum of only 31,898. Furthermore, the median number of sequences per sample was 953. As shown in Table 3, the branch length means differ for NJ and UPGMA. The UPGMA mean branch length is 1585, whereas the NJ branch length is around 600. The variance is higher for neighbor joining, for the 12S marker, consistent with the other markers. For the NJ tree topology, k = 2 appears to be the number of clusters, whereas for UPGMA, k = 3 is apparent for the number of clusters in terms of 12S sequences identified.

In Figure S12, the PCA for the 12S DNA sequences that were recovered from the L.A. River sediment samples is given. The first two principal components capture about 63% of the variation in the data. Samples appeared similar in this assay, except for the sample from high on PC 2 in the Elysian Valley that contained a high relative abundance of salmon sequences, which appeared to be an error. In that case, since the taxon is too rare among samples, it could be excluded from the analysis because it might be an error or was unlikely to be relevant to many individuals in the population. Figure S13 shows the PCA plot for the 12S samples color coded by the best PAM clustering, which was k = 5, with the highest average silhouette width. A total of 79 out of 90 samples fall into the first cluster, shown in black. The second cluster is mostly made up of Sepulveda Dam sediment DNA samples. The first and third clusters were similar to one another. The fifth cluster, in light blue, is made up of a single sample from Long Beach.

The observed plant alpha diversity for each of the L.A. River sites is plotted in Figure 5. The median number of assigned sequences per sample was relatively low for the plant ITS assay at 9642, although it was not the lowest of all markers. Nevertheless, the number of sequences per taxa had a high maximum at 238,793. As shown in Table 3, the branch length means were similar for NJ and UPGMA, and the variance is about 250,000 units higher for neighbor joining, with respect to the PITS marker. For both tree topologies, k = 4 is reflected for the number of clusters in terms of plant sequences identified.

In Figure S14, it is possible to view the PCA for the plant DNA sequences that were recovered from the L.A. River sediment samples. These data are interesting in terms of assessing ecosystem diversity and nitrogen cycling. The first two principal components capture about 34% of the variation in the data. One of the samples from Elysian Valley is high on PC4 due to a high abundance of *Paspalum distictum* sequences. This is a knotgrass found in most of the southern US and the Pacific northwest, where it is native but can become weedy [75]. *Paspalum* plays a role in wetland restoration, since it tolerates waterlogged and saline environments, as well as providing food for deer [75]. Samples from Arroyo Seco are high on PC 3 based on the differential abundance of *Alnus rhombifolia* sequences. Interestingly, most of the Alnus sequences were derived from a Tujunga Wash sample. White alders are native to streamside habitats in the US [76]. Alders have been shown to

be key to nitrogen cycling in riparian environments, since they form an association with *Frankia* bacteria. For that reason, they are better at colonizing disturbed habitats [76].

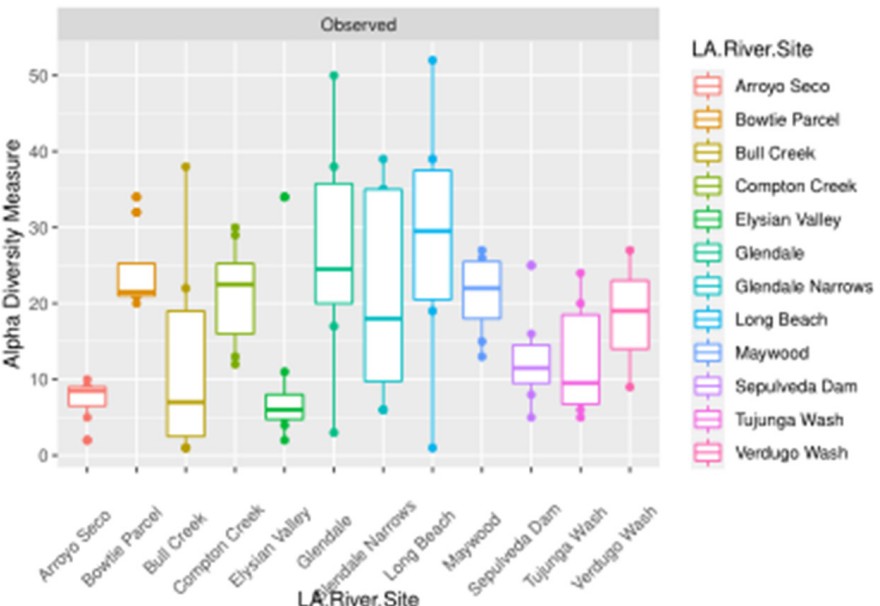

**Figure 5.** The observed alpha diversity of plant species is depicted in boxplots. This figure answers the question: Which site had the highest number of plant species detected overall? Note that the highest observed alpha diversity tended to be in Glendale, Glendale Narrows, and Long Beach. Again, there is evidence of overdispersion, especially for the Bull Creek, Glendale, and Long Beach samples.

The main factor that separates samples on PC 2 is the abundance of willow species, especially in Bull Creek, Bowtie Parcel, and Arroyo Seco. Most of the Salix sequences were derived from two samples from Arroyo Seco. Figure 6 shows the PCA for the plant sequences, color coded by the best PAM clustering. The best PAM clustering for the FITS markers was k = 4. The model with four clusters had the highest average silhouette width. The second cluster, shown in red, is composed of Arroyo Seco and Bull Creek. The third cluster consists of sediment samples from Compton Creek-, Sepulveda Dam-, and Glendale-adjacent sites. The fourth cluster, in blue, is made up of Arroyo Seco samples. The first cluster is made up of a mixture of all other samples, which were similar to one another, shown in black.

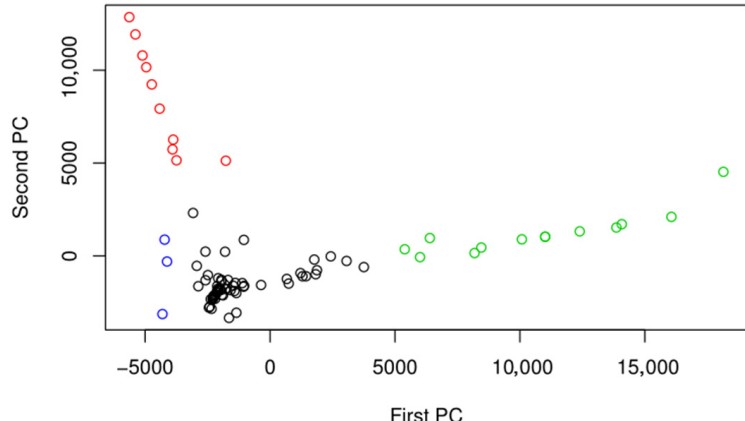

**Figure 6.** PCA for identified plant sequences from the PITS marker by sample is presented, color coded by the best PAM clustering. Each cluster was color coded according to overall similarity of the plant populations.

## 4. Discussion

This study has investigated the associations between microorganisms and environmental conditions including soft bottoms versus concrete bottoms, the degree of urbanization, and proximity to a water treatment plant. The physical distance between samples appears to be mirrored by the genetic distance, based on the evidence from PCA with PAM clustering for the 18S markers. Matsuoka et al. found similar results along a river network in Japan in 2019, where they found that fungal DNA assemblages had a spatial structure and samples that were closer to one another tended to be more similar. Overall, our results agree with the numerous studies of urban, eutrophic, and brackish freshwater bodies since proteobacteria, bacteroidetes, firmicutes, cyanobacteria, chloroflexi, actinobacteria, and acidobacteria were all well-represented [77–80]. The elevated presence of Verrucomicrobia and Gammaproteobacteria aligned more with the brackish metagenome [80]. The ostracods detected in high abundance are not known indicator species for heavy metal contamination [81].

In Glendale Narrows, downstream from water reclamation plants, there were abundant cyanobacteria and algae sequences. Eutrophication can lead to hypoxic conditions; since hypoxia can be fatal to fish, this may partly explain the low 12S diversity. The greatest social costs associated with irrigating using reclaimed water are the costs to recreation and the risks to human health due to the potential for the presence of hazardous substances [82]. However, at Glendale Narrows, indicator species for both low nutrient environments and ammonia-abundance were also present. A potential explanation for this is the high abundance of plant species at Glendale Narrows, which assimilate nitrogen. Microbes with nutrient cycling capabilities, such as nitrogen reduction or nitrogen fixation, have been known to be associated with plant growth promotion or may be associated with toxicity. Nevertheless, our results do not agree with Francis et al., 2012, where plant species diversity was expected to decrease in urban environments compared to rural environments [83].

Eukaryotic microbes in the rootzone, such as *Basidiomycota* and *Ascomycota* may help plants with phosphorus solubilization but may be pathogenic to plants or humans. Organisms such as these fungi, which promote phosphorus mineralization, have received less attention over the years [84], although they play important roles in nutrient cycling. Fungi such as *Pleurotus* have been shown to mycoremediate contamination with *E. coli* [85]. The results indicate that L.A. River biome is rich with *Ascomycota* beyond the expected proportion for freshwater bodies, including rivers. *Penicillium* sp. are known to bioaccumulate arsenic and cadmium and are thus mycoremediators of metals [86].

Nitrogen cycling was explained through the differential abundance of ammonia oxidizing archaea; the complete ammonia-oxidizers, *Nitrospira* sp.; nitrate-reducing bacteria, *Marmoricola* sp.; and nitrogen-fixing bacteria, *Devosia* sp., were differentially abundant at soft-bottom sites ($p$ adj < 0.002). The proposed nitrogen cycle for soft-bottom conditions is shown in Figure 7. Ammonia-oxidizing archaea were represented by more than one species. This result partly disagrees with the findings by Cai et al. [87], since ammonia oxidizing archaea were more represented. However, some *Nitrospira* bacteria are complete ammonia oxidizers, so they may be equally important. Interestingly, the results from a recent study indicated that nitrogen pollution in river sediments also contributed to bacteria community shifts [78]. In contrast, the differential abundance of several *cyanobacteria* and other anoxygenic phototrophs was associated with the concrete-bottom sites, which suggested the accumulation of excess nitrogen. *Desulfomicrobium* may play a part in nitrate reduction in concrete environments but conserves more nutrition [66] and is sulfate-dependent [65]. Since denitrification generally requires substrate that is made under aerobic conditions [88], it makes sense that denitrifying bacteria were not as abundant in the concrete environments. Clostridia are indicator species for fecal contamination and sewage [89]. In regard to the reproductive pathogens, as Hervé et al. noted, street gutters are important in the dispersal of putative pathogens from anthropogenic waste [90] and bioremediating species.

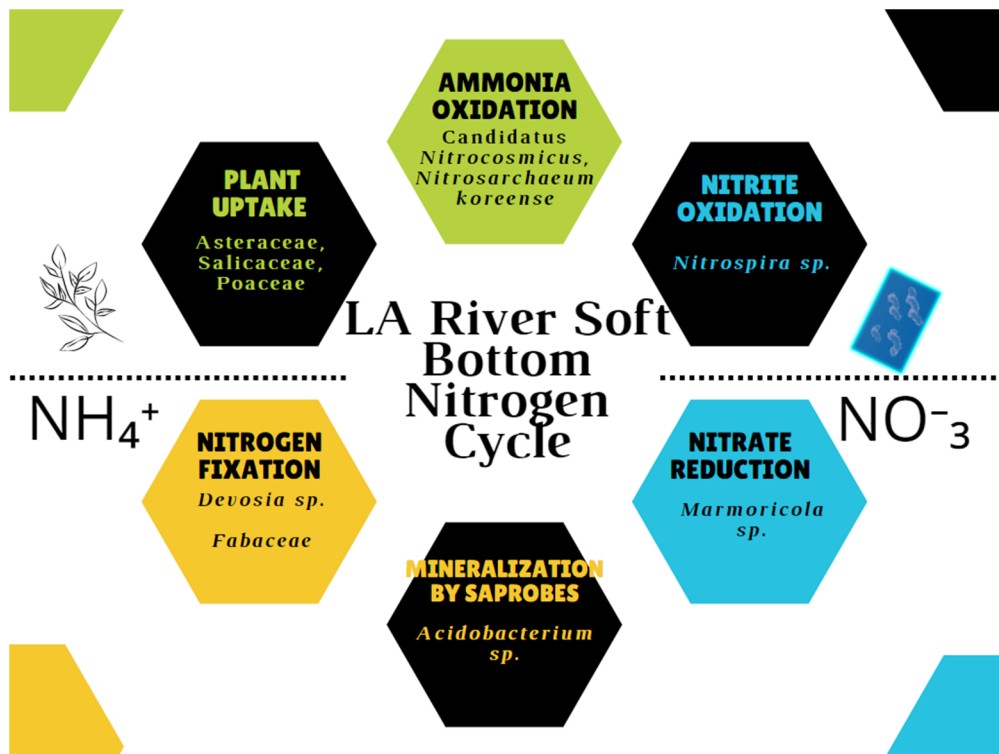

**Figure 7.** Proposed nitrogen cycle for the L.A. River soft-bottom conditions.

The diversity of cyanobacterial species observed indicated health within the cyanobacteria community. As Stal noted in 2007, *cyanobacteria* are involved in two essential biogeochemical processes on Earth, since they capture both $CO_2$ and $N_2$ [88]. *Cyanobacteria* have been known to colonize hostile environments [91] and to produce toxins that bring health risks to the public, such as liver damage, eye irritation, vomiting, and death [92]. However, only 1–2 species of algae were highly represented, which is not an indicator of health for the ecosystem. In a freshwater study by Wang et al., elevated cyanobacteria were associated with bacterioplankton, whereas algae were associated with zooplankton [93]. The heterogeneity and diversity of algae is tied to ecosystem services [94]. According to the Southern California Coastal Water Research Project, *Cladophora* algae support the habitat of wading shorebirds [95]. Treating the underlying anaerobic conditions could promote algal and fish diversity.

The soft-bottom sites tended to be represented by differential abundance of aerobes, whereas the concrete-associated species tended to be alkaliphilic, saliniphilic, calciphilic, sulfate-dependent, and anaerobic. The presence of halophiles is a good indicator of salinity problems. The differential abundance of *Proteobacteria* was associated with soft-bottom sites, and there was an apparent balance in the abundance of organisms responsible for nitrogen cycling.

In recent years, the city of Los Angeles has been reluctant to move toward a soft-bottom channel restoration, since it would necessitate a widening of the channel, which would potentially affect landowners and other infrastructure. Furthermore, although some activists have favored riparian plantings, this also has the potential to slow the flow of water. As the river was channelized in order to decrease flooding risk and efficiently carry away water, the introduction of a vegetative buffer would likely require a widening of the river, and possibly the river's overall footprint. As Levi et al. pointed out, channel restoration benefits appear to be smaller when spread across a larger area [96]. Therefore, this type of effort may be most impactful when applied to the urban stretches that would benefit most from the intervention.

Based on the plant diversity analysis, it was indicated that Maywood had high sequence abundances of weeds such as Datura, Atriplex, Oxalis, and Chenopodium, as well as a high abundance of toxic cyanobacteria based on the factor analysis; therefore, Maywood could benefit from the planting of perennial foliage that could also remediate air pollution [97]. According to Liu et al., air pollutants, including particulate matter, nitrous oxide, and carbon monoxide, also influence microbial and fungal communities [98]. Indications tended to suggest that sonicating devices at Maywood and Glendale Narrows for the control of *Cyanobacteria* should be considered, as well as perennial vegetative buffers in Maywood to combat noxious *Datura* plant species and toxic *cyanobacteria* blooms. Interestingly, Maywood samples had differentially abundant *Tetradesmus* sp., including *T. obliquus*, which is a phosphorus accumulator and produces valuable lipids for biodiesel [42]. *T. obliquus* may also be used for animal feed; it is known to be rich in amino acids, including the essential amino acid leucine, with a low bioaccumulation of metals [42].

A surprising result is that some sites along the L.A. River were more diverse with plant life than rural Arroyo Seco, especially Bowtie Parcel, Glendale, Long Beach, and Maywood, based on observed alpha diversity. This is most likely due to the landscape plantings of exotic species near Glendale, coastal species at Long beach, and a diverse panel of weed sequences that were identified at Maywood. Plants prevent erosion and create habitats for birds, mammals, invertebrates, amphibians, and reptiles. Plants also help balance nitrogen cycling and can provide a buffer by absorbing some of the nutrients involved in eutrophication. Native plants are useful for bioremediation, soil stabilization, habitat restoration, and as a replacement for invasive species. Native Californian species would also create habitat for wildlife, including birds, insects, and many pollinators. Suggestions for plants for the L.A. River embankment are given in Table 7.

**Table 7.** Some native Californian plant suggestions for the L.A. River embankment.

| Botanical Name | Common Name | Category | Environment |
|---|---|---|---|
| *Artemesia douglasiana* | Douglas' sagewort | Smaller shrubs and perennials | normal, moist, or saturated soils |
| *Carex praegracilis* | field sedge | Smaller shrubs and perennials | normal, moist, or saturated soils |
| *Eleocharis macrostachya* | common spikerush | Smaller shrubs and perennials | normal, moist, or saturated soils |
| *Equisetum hyemale* | horsetail | Smaller shrubs and perennials | normal, moist, or saturated soils |
| *Juncus patens* | common rush | Smaller shrubs and perennials | normal, moist, or saturated soils |
| *Ribes aureum var. gracillimum* | golden currant | Smaller shrubs and perennials | normal, moist, or saturated soils |
| *Rosa californica* | California wildrose | Smaller shrubs and perennials | normal, moist, or saturated soils |
| *Verbena lasiostachys* | vervain | Smaller shrubs and perennials | normal, moist, or saturated soils |
| *Acer negundo* | box elder | Larger shrubs and trees | normal, moist, or saturated soils |
| *Acer rhombifolia* | white alder | Larger shrubs and trees | normal, moist, or saturated soils |
| *Baccharis salicifolia* | mulefat | Larger shrubs and trees | normal, moist, or saturated soils |
| *Juglans californica* | black walnut | Larger shrubs and trees | normal, moist, or saturated soils |
| *Platanus racemosa* | California sycamore | Larger shrubs and trees | normal, moist, or saturated soils |
| *Populus fremontii* | Fremont cottonwood | Larger shrubs and trees | normal, moist, or saturated soils |
| *Salix laevigata* | red willow | Larger shrubs and trees | normal, moist, or saturated soils |
| *Salix lasiolepis* | arroyo willow | Larger shrubs and trees | normal, moist, or saturated soils |
| *Sambucus mexicana* | blue elderberry | Larger shrubs and trees | normal, moist, or saturated soils |
| *Artemesia californica* | California sagebrush | Smaller shrubs and perennials | riparian banks, not saturated |
| *Asclepias fasiculata* | narrow leaf milkweed | Smaller shrubs and perennials | riparian banks, not saturated |
| *Encelia californica* | bush sunflower | Smaller shrubs and perennials | riparian banks, not saturated |
| *Eriogonum fasciculatum* | California buckwheat | Smaller shrubs and perennials | riparian banks, not saturated |
| *Lotus scoparius* | deerweed | Smaller shrubs and perennials | riparian banks, not saturated |
| *Salvia apiana* | white sage | Smaller shrubs and perennials | riparian banks, not saturated |
| *Salvia clevelandii* | Cleveland sage | Smaller shrubs and perennials | riparian banks, not saturated |
| *Salvia mellifera* | black sage | Smaller shrubs and perennials | riparian banks, not saturated |
| *Baccharis pilularis* | coyote brush | Larger shrubs and trees | riparian banks, not saturated |

**Table 7.** *Cont.*

| Botanical Name | Common Name | Category | Environment |
|---|---|---|---|
| *Ceanothus* spp. | California lilac | Larger shrubs and trees | riparian banks, not saturated |
| *Heteromeles arbutifolia* | toyon | Larger shrubs and trees | riparian banks, not saturated |
| *Juglans californica* | California walnut | Larger shrubs and trees | riparian banks, not saturated |
| *Manzanita* spp. | | Larger shrubs and trees | riparian banks, not saturated |
| *Malosma laurina* | laurel sumac | Larger shrubs and trees | riparian banks, not saturated |
| *Platanus racemosa* | California sycamore | Larger shrubs and trees | riparian banks, not saturated |
| *Rhus integrifolia* | lemonade berry | Larger shrubs and trees | riparian banks, not saturated |
| *Sambucus mexicana* | blue elderberry | Larger shrubs and trees | riparian banks, not saturated |
| *Quercus agrifolia* | coast live oak | Larger shrubs and trees | riparian banks, not saturated |

## 5. Conclusions

Further research should consider the efficacy of sonicating devices at Maywood and Glendale Narrows for the control of *cyanobacteria* [99]. There were poorly characterized microbes and arthropods identified in this study that may present an opportunity for further investigation. These include a possible new species of *Capniodales* sooty mold in the submerged samples, little known *Chironomidae* lake flies in the Glendale Narrows sample, *Desulfomicrobia* in concrete environments, elusive *Eustigmatophyaceae* in Maywood, and unstudied *Verrucomicrobia* and *Flavobacter* in Glendale Narrows. Arroyo Seco and Maywood, which are geographically connected, present an interesting junction of the L.A. River to investigate Ascomycetes and sequence them to a deeper level. This is one of the first attempts to characterize the metagenome of the L.A. River. The diversity and interaction of the bacterial communities with plants and other organisms warrants more attention. The outcomes appear to involve interactions between environmental factors. Further research should consider the functional analysis of similar associations.

**Supplementary Materials:** The following supporting information can be downloaded at: https://www.mdpi.com/article/10.3390/d15070823/s1.

**Author Contributions:** Conceptualization, S.S.; methodology, S.S.; formal analysis, S.S.; investigation, S.S., D.M. and R.S.; writing—original draft preparation, S.S.; writing—review and editing, A.E.T., D.M., G.P., S.B., R.S., K.P. and J.F.; visualization, S.S.; supervision, G.P., A.E.T. and S.B. All authors have read and agreed to the published version of the manuscript.

**Funding:** The authors acknowledge support from the University of California CaleDNA Program who kindly provided us with data prior to publication.

**Institutional Review Board Statement:** Not applicable.

**Data Availability Statement:** Research data for the L.A. River Round 1 Project are available from CaleDNA at: https://data.ucedna.com/research_projects/los-angeles-river-round-1/pages/introduction, accessed on 3 April 2023.

**Acknowledgments:** Thank you to John Creedon and Elias Tarver for technical writing assistance.

**Conflicts of Interest:** The authors declare no conflict of interest.

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
