# Peer review of "The Community Structure of eDNA in the Los Angeles River Reveals an Altered Nitrogen Cycle at Impervious Sites"

_diversity, doi:10.3390/d15070823_

Round 1

Reviewer 1 Report

The manuscript, titled " Partitional Clustering and Differential Abundance Analysis Reveal the Community Structure of eDNA in the Los Angeles River " presents a novel idea of using environmental DNA to reveal the community structure including bacteria, plants, fungi, fish, and invertebrates of the Los Angeles River. This is a challenging project, and the study also used a variety of statistical analysis methods. However, the manuscript does not complete the expected research assumptions well as a whole, for the following reasons:

l  The research background of the manuscript is insufficient, and the background of eDNA related cases and the status quo of community structure in Los Angeles River are lacking

l  The study lacks important data information, such as the results of species information based on eDNA identification of different community structures in the Los Angeles River

l  The overall logic of the manuscript is confused, large amounts of data are simply presented without logical analysis, and the method is not matched with the results. The main research purpose is not clearly expressed, and the title does not match the content of the manuscript.

l  There are many formatting errors in the manuscript, such as repeated text (line 62-67) and non-standard references, figures and tables.

    Based on the above, I cannot accept the present manuscript.

Author Response

Thank you for your comments, which have improved this manuscript.  The title was changed in response to your comment and in order to appeal more to the interest groups you called out.  The logic of the manuscript was improved by using the overarching theme given in the title, which was about nitrogen cycling, as the link that ties together the information about plants and bacteria.  

There was no status quo for the LA River, since this was the first large scale study.  The family, genus, and species information are given for a large number of taxa identified in the study.  

The captions to the figures have been included where missing and formatting errors have been corrected, as well as making several sections more concise or clear.

Reviewer 2 Report

Dear authors, 

The results of your study could be indeed of high interest for conservationists, managers of the river and the scientists as well. However, overall, it seemed to me hard to follow the issues exposed. The results and discussions seem a little bit confusing, not following a red thread. You talked a little bit about everything but not in a structured way. Please, try to give more fluency to your ideas. I attach the the manuscript, where I highlighted some phrases not clear for me.

Author Response

Thank you for your suggestions, which have been incorporated into the text. Several formatting errors and typos have been fixed and captions have been added where missing. Language has been revised in several passages where indicated in your markup. The overarching theme of nitrogen cycling was used to tie together the plant and bacterial information thematically. Emphasizing this connection also highlighted the ecological relevance.  Differential abundance was defined and this overused term was replaced in some instances for clarity and variation.  

For the PCA PAM figures, the percent variation is given in the body of the text and in the PCA variations given in the supplemental materials.  Tables 5 and 6 were edited as you have indicated.  Figure 4 was explained in detail in the text (added).

The relevance of ostracods and bloodworms as positive ecological indicator species was noted on page 20.

The language in the discussion was updated, and it should be noted that hypoxia was not directly measured but the high abundance of Cyanobacteria is evidence. 

Thank you for your comments, which have improved the text.